# Pressure–Volume Loop Analysis of Voiding Workload: An Application in Trans-Vaginal Mesh-Repaired Pelvic Organ Prolapse Patients

**DOI:** 10.3390/bioengineering10070853

**Published:** 2023-07-19

**Authors:** Hui-Hsuan Lau, Cheng-Yuan Lai, Ming-Chun Hsieh, Hsien-Yu Peng, Dylan Chou, Tsung-Hsien Su, Jie-Jen Lee, Tzer-Bin Lin

**Affiliations:** 1Division of Urogynecology, Department of Obstetrics and Gynecology, Mackay Memorial Hospital, Taipei 11031, Taiwan; 2Department of Nursing, Mackay Junior College of Medicine, Nursing, and Management, Taipei 11031, Taiwan; 3Department of Medicine, Mackay Medical College, New Taipei 25244, Taiwan; 4Institute of Biomedical Sciences, Mackay Medical College, New Taipei 25244, Taiwan; 5Department of Surgery, Mackay Memorial Hospital, Taipei 10449, Taiwan; 6Institute of Translational Medicine and New Drug Development, College of Medicine, China Medical University, Taichung 40402, Taiwan; 7Cell Physiology and Molecular Image Research Center, Wan Fang Hospital, Taipei Medical University, Taipei 11689, Taiwan; 8Department of Physiology, School of Medicine, College of Medicine, Taipei Medical University, No. 250 Wu-Shin Street, Taipei 11031, Taiwan

**Keywords:** pressure–volume analysis, thermodynamics, urethral resistance, pelvic floor reconstruction, trans-vaginal mesh repair

## Abstract

Although trans-vaginal mesh (TVM) offers a successful anatomical reconstruction and can subjectively relieve symptoms/signs in pelvic organ prolapse (POP) patients, its objective benefits to the voiding function of the bladder have not been well established. In this study, we investigated the therapeutic advantage of TVM on bladder function by focusing on the thermodynamic workload of voiding. The histories of 31 POP patients who underwent TVM repair were retrospectively reviewed. Cystometry and pressure volume analysis (PVA) of the patients performed before and after the operation were analyzed. TVM postoperatively decreased the mean voiding resistance (mRv, *p* < 0.05, N = 31), reduced the mean and peak voiding pressure (mPv, *p* < 0.05 and pPv, *p* < 0.01, both N = 31), and elevated the mean flow rate (mFv, *p* < 0.05, N = 31) of voiding. While displaying an insignificant effect on the voided volume (Vv, *p* < 0.05, N = 31), TVM significantly shortened the voiding time (Tv, *p* < 0.05, N = 31). TVM postoperatively decreased the loop-enclosed area (Apv, *p* < 0.05, N = 31) in the PVA, indicating that TVM lessened the workload of voiding. Moreover, in 21 patients who displayed postvoiding urine retention before the operation, TVM decreased the residual volume (Vr, *p* < 0.01, N = 21). Collectively, our results reveal that TVM postoperatively lessened the workload of bladder voiding by diminishing voiding resistance, which reduced the pressure gradient required for driving urine flow.

Clinical Trial Registration: ClinicalTrials.gov (NCT05682989)

## 1. Introduction

The prevalence of pelvic organ prolapse (POP), in which organ(s) protrude beyond their anatomical confines in the pelvic cavity, is increasing as life span continues to advance because POP most commonly affects women older than 70 years [1,2]. In addition to bowel and sexual disorders, many POP patients suffer from voiding dysfunctions, such as straining to void, incomplete bladder emptying, or a weak urine stream [3], which could be detrimental because lasting urine retention subsequent to inadequate bladder emptying could result in frequent and/or recurrent urinary tract infections [4]. Therefore, therapies for POP have been developed not only for treating bowel or sexual dysfunctions but also for remedying voiding deficits [5].

Among therapeutic options for symptomatic POP, including consulting, observation, physical therapy, vaginal pessary, and surgical intervention [5], trans-vaginal mesh (TVM), which is designed to correct anatomical abnormalities and thereby restore pelvic floor function [6], is a minimally invasive surgery for POP [7]. Although TVM can cause bladder damage, lower urinary tract symptoms, pelvic pain, dyspareunia [8,9], and mesh erosion [10,11] as side effects, the USA Food and Drug Administration issued a notification regarding the long-term safety of TVM repair in 2016 [12]. Some TVM kits [13,14,15] are still viable options in Asia and continental Europe for treating symptomatic POP because they offer successful anatomical reconstruction and satisfying subjective outcomes [16]. Nevertheless, the objective therapeutic benefits of TVM, particularly its functional effectiveness in voiding dysfunctions, have been scarcely investigated.

It is well-recognized that the obstruction in the bladder outlet [17] caused by kinking and/or compression of the urethra [18] underlies the development of voiding dysfunctions in POP patients [17,18,19]. Since urethral obstruction manifests itself as enhanced outlet resistance during urine emission, [20] and a very recent study has demonstrated that enhanced outlet resistance is associated with an increased thermodynamic workload of the bladder emptying [21], the current study was designed to explore whether TVM lessens the workload of bladder voiding by reducing the voiding resistance. For this purpose, the impact of TVM on the thermodynamic work and urethral resistance of voiding in POP patients were analyzed by retrospectively comparing the pre and postoperative urodynamic investigations. Moreover, to clarify possible mechanisms involved in the effects caused by TVM, workload/resistance-associated parameters, namely, the voiding pressure, flow rate, voided volume, and voiding time, were also measured and analyzed.

## 2. Patients and Methods

### 2.1. Study Design

This study was reviewed and approved by the ethics committee of Mackay Memorial Hospital, Taipei, Taiwan (22MMHIS361e; 2022/12/08), and the protocols were registered in ClinicalTrials.gov (NCT05682989). A history of POP patients with concurrent objective and/or subjective voiding dysfunctions who underwent primary TVM surgery for symptomatic POP ≥ stage II (pelvic organ prolapse quantification system) in Mackay Memorial Hospital from January 2007 to December 2022 was analyzed. Patients with a history of pelvic radiation, vesico-vaginal, recto-vaginal, or urethra-vaginal fistula and those who were unable to be followed up were excluded from this study. The primary outcome assayed in this study was the voiding resistance, and the secondary outcomes were the mean and peak voiding pressure, voiding flow, voided volume, voiding time, loop-enclosed area, and residual volume.

### 2.2. Surgery

The patient received cefmetazole (1 g, intravenous) as prophylactic antibiotics before skin incision; if indicated, a vaginal hysterectomy was performed. Trans-vaginal mesh repair was performed using a surgical mesh kit (Surelift^®^, Neomedic International, Barcelona, Spain). First, at the anterior vaginal wall, a vertical incision was made from the level of the bladder neck to the cervix or vaginal vault. Following lateral dissection to the level of the arcus tendineus and then caudal to the bilateral ischial spines, a full-thickness hydrodissection to the vesico-vaginal space was identified. Two centers of the six mesh arms were cut off as modifications for all patients. Two posterior arms were fixed to the sacrospinous ligaments (approximately 1.5 cm medial from the ischial spines) through the anchors with polypropylene thread by an applicator (Anchosure System^®^, AFS MEDICAL, Teesdorf, Austria). After the posterior arms were in place, the other two anterior arms were penetrated through the obturator foramina. Then, the mesh was adjusted to be tension-free to ensure that there was no folding in the mesh. To prevent mesh migration that possibly causes recurrent cystocele, sutures were placed beneath the bladder neck and near the vaginal vault. Finally, the excess part of the anterior arms was cut off at the level of the skin, and the vaginal epithelium was closed.

### 2.3. Cystometry Investigation

Protocols for cystometry investigations complied with the guidelines of the International Continence Society (ICS) [22]. In brief, a multichannel urodynamic study in which warm saline (37 °C) was continuously infused into the bladder of patients with an infusion rate of 80 mL/min was recorded online (MMS UD-200, Medical Measurement System, Enschede, the Netherlands) and analyzed offline (Biopac MP36, Biopac Systems, Santa Barbra, CA, USA) using computer systems. The detrusor pressure (Figure 1A Pdet), vesical pressure (Pves), abdominal pressure (Pabd), urethral flow (Flow), infused volume (Vinf), voided volume (Vvod), and intravesical volume (Vive) were recorded online, and the mean voiding pressure (Figure 1B mPv; the mean Pdet during fluid emission), voiding time (Tv; the duration of fluid emission), voided volume (Vv; the volume of emitted fluid), mean voiding flow (mFv; calculated by Vv/Tv), and mean voiding resistance (mRv; calculated by mPv/(Vv/Tv)) were analyzed offline.

### 2.4. Pressure–Volume Analysis

Derived from the cystemetry, the pressure–volume analysis (PVA) of voiding cycles was established by plotting Pdet against Vive (Figure 1C) [23,24,25]. The residual volume (Vr) of voiding was calculated by the volume difference between the infused volume and voided volume of a voiding cycle (i.e., the volume difference between the right and the left bolder of the loop). The trajectory-enclosed area (Apv) was analyzed using an image-processing program (ImageJ, LOCI, Madison, WI, USA). Although cough and Valsalva tests markedly interfere with Pves and Pabd, a previous publication [26] and our data (Figure 1A,B) demonstrated that these tests displayed little effect on Pdet in cystometry. In addition, our data revealed that the cough and Valsalva maneuvers trivially impacted the trajectory of the PVA (Figure 1C). Moreover, cough and Valsalva tests were carried out during urine storage, which would barely affect the voiding dynamics. Therefore, cystometry data of TVM patients who underwent stress tests (coughs and/or Valsalva) were used in this study.

### 2.5. Statistical Analysis

Baseline characteristics, including the age of patients and the days that cystometry was performed before and after the operation, were summarized using descriptive statistics. Statistical data are expressed as the mean ± SEM. Differences in values between groups were assessed using paired Student’s *t*-tests. A significant difference was set at *p* < 0.05.

## 3. Results

### 3.1. Baseline Data of the Patient

Urodynamic data obtained from 31 POP patients with a mean age of 66.15 ± 1.57 years old were reviewed and analyzed in the current study. All patients showed at least stage II prolapse (POP quantification system). Urodynamic evaluations were conducted at a mean of 69.32 ± 15.42 days before and 188.00 ± 47.97 days after TVM surgery.

### 3.2. TVM Postoperatively Diminishes the Voiding Resistance

Because voiding dysfunction in POP patients is presumed to be resulted from kinking and/or compression-enhanced voiding resistance in the bladder outlet [18,20], whether TVM diminishes voiding resistance in POP patients was investigated as the first part of this study. Representative urodynamic investigations carried out pre (Figure 2A PRE) and postoperatively (Figure 2B POST) demonstrated that when compared with the preoperative control, TVM consistently decreased mean voiding resistance (mRv) in most patients (Figure 2C; 26 out of 31 patients) and significantly decreased the mean mRv of the patient (*p* < 0.05 vs. PRE, N = 31). This result indicated that TVM postoperatively diminished the voiding resistance of POP patients.

### 3.3. TVM Postoperatively Decreases the Voiding Pressure

Considering that the voiding resistance (Rv) is defined by dividing the voiding pressure (Pv) by the flow rate (Fv; Rv = Pv/Fv), we first investigated whether the TVM-diminished resistance is associated with changes in the voiding pressure and/or flow rate by inspecting whether the TVM reduces the pressure developed for voiding. When compared with the preoperative control, TVM decreased both the mean voiding pressure (mPv; Figure 3A, 21 out of 31 patients) and peak voiding pressure (pPv; Figure 3B, 25 out of 31 patients) in most patients and significantly decreased both the mean mPv and pPv of the patient (*p* < 0.05 and *p* < 0.01, vs. PRE, respectively, both N = 31). These results indicated that the bladder of POP patients develops a lower pressure during voiding after TVM.

### 3.4. TVM Postoperatively Elevates the Flow Rate

On the other hand, we assayed whether the TVM-diminished resistance was associated with a modified flow rate during voiding. When compared with the preoperative control, we found that TVM increased the mean flow rate of voiding (mFv) in most patients (Figure 3C; 23 out of 31 patients) and significantly increased the mean mFv of the patient (*p* < 0.05, vs. PRE, N = 31). This result indicated that TVM-diminished resistance was accompanied by an elevated emission rate. Together with the above result, these findings reveal that the TVM-diminished voiding resistance is associated with a decreased voiding pressure and elevated flow rate of urine emission.

### 3.5. TVM Postoperatively Shortens the Voiding Time

Next, to further analyze the underlying causes of the TVM-elevated flow rate of voiding, the impact of TVM on the voided volume (Vv) and voiding time (Tv) was analyzed, as the flow rate was calculated by dividing Vv by Tv (Fv = Vv/Tv). We observed that when compared with the preoperative control, TVM neither resulted in a consistent trend of Vv change in patients (Figure 4A) nor significantly affected the mean Vv of the patient (*p* > 0.05, vs. PRE, N = 31). In contrast, when compared with the preoperative control, TVM postoperatively decreased Tv in most patients (Figure 4B; 22 out of 31 patients) and significantly decreased the mean Tv of the patient (*p* < 0.05 vs. PRE, N = 31). Collectively, these findings revealed that the TVM-elevated emission rate could be attributed to a shortened voiding time with a negligible effect on the modified voided volume.

### 3.6. TVM Postoperatively Lessens the Voiding Workload

We have observed that TVM effectively diminished the voiding resistance, and patients postoperatively developed decreased bladder pressure but drove urine with a higher emission rate. We wondered whether these findings collectively reveal that TVM lessens the workload of voiding in POP patients. For this purpose, we established pressure–volume analyses (PVAs) because the trajectory-enclosed area (Apv) in the PVA is presumed to represent the work performed in a voiding cycle [24,25]. Illustrative PVA demonstrated that without markedly affecting the level of the bottom border or the volume difference between the left and right borders, TVM postoperatively decreased the Apv as it depressed the top border when compared with the preoperative control (Figure 5A PRE and Figure 5B POST). The Apv decrement was confirmed by the summarized data demonstrating that when compared to the preoperative control, TVM postoperatively decreased Apv in most patients (Figure 5C; 23 out of 31 patients) and significantly decreased the mean Apv of the patient (*p* < 0.05 vs. PRE, N = 31).

### 3.7. TVM Postoperatively Ameliorates Urine Retention

Finally, we wondered whether the TVM-lessened workload of voiding could result in impaired voiding efficacy in POP patients. For this purpose, we investigated whether TVM exaggerates or causes urine retention in patients. In 21 out of the 31 patients, their preoperative PVA displayed more infused volume than voided volume (Figure 5A PRE), indicating an amount of postvoid residual volume (Vr) in the bladder. Moreover, although the volume difference between the right and left borders was not evidently affected, TVM markedly shifted the loop to the left side, indicating that the Vr was decreased postoperatively (Figure 5B POST). The decrement in Vr was confirmed by the summarized data showing that when compared with the preoperative control, TVM decreased Vr in most of the 21 patients (Figure 5D; 17 out of 21 patients) and significantly decreased the mean Vr of the 21 patients (*p* < 0.01 vs. PRE, N = 21). On the other hand, 10 out of the 31 patients who were free of Vr before the operation also completely emptied their bladder postoperatively. These results reveal that TVM neither exaggerated nor brought about urine retention in POP patients, suggesting that TVM-lessened voiding work was not accompanied by impaired voiding efficacy; instead, it was associated with increased voiding efficacy.

## 4. Discussion

In addition to bowel and/or sexual disorders, many POP patients suffer from voiding dysfunctions such as straining to void, incomplete bladder emptying, or a weak urine stream [2]. Because kinking- and/or compression-enhanced outlet resistance is recognized to underlie the development of voiding difficulties in POP patients [17,19], we explored in this study the potential benefit of TVM to the voiding function by specifically focusing on outlet resistance-associated voiding work. The results of cystometry demonstrate that TVM consistently and significantly diminished the urethral resistance of patients, suggesting that TVM benefits POP patients by relieving the kinking/compression-increased outlet resistance. In addition, we propose that the diminished voiding resistance lessened the pressure gradient requisite for driving urine flow and thereby reduced the voiding work, prompting urine emission to increase voiding efficacy.

### 4.1. TVM Lessens the Thermodynamic Work of Bladder Voiding

Our conclusion is based on the findings. First, our data revealed that TVM-decreased Apv was associated with reduced voiding pressure but statistically unaffected voided volume. This finding is supported by PVA diagrams showing that TVM postoperatively decreased the level of the top border (which represented the voiding pressure) without markedly affecting the intercept between the left and right borders (which represented the voided volume) of the loop. Considering that Apv is the integral of the voiding pressure with the voided volume, these findings collectively imply that the TVM-decreased Apv was mainly attributed to the depressed voiding pressure. Furthermore, since the voiding resistance and voiding pressure are proportional, our urodynamic measurements revealed that the TVM-diminished resistance was significantly accompanied by decreased voiding pressure. These findings imply that the bladder developed a lower voiding pressure as the pressure gradient required for propelling urine flow was decreased in response to the TVM-diminished voiding resistance.

Together with the fact that Apv is presumed to represent the thermodynamic work performed in a voiding cycle [24,25], these several findings suggest that with an unaffected voided volume, the TVM-diminished resistance reduced the developed pressure necessary for driving an adequate urine flow and thereby lessened the workload of bladder voiding. Considering that studies have demonstrated that a long-term enhanced workload is a risk factor leading to uncompensatory bladder functions [27], these findings collectively suggest that TVM benefits POP patients as it relieves the workload of bladder voiding.

### 4.2. TVM Postoperatively Increases the Voiding Efficacy of the Bladder

Questions that need to be clarified are whether TVM-lessened voiding work could impair voiding efficacy by diminishing urine flow or exaggerating/causing urine retention. Nevertheless, these proposals do not seem to be the case because even though the peak and mean voiding pressures were both decreased, the TVM-associated Apv decrement was accompanied by a consistently and significantly enhanced flow rate of voiding, i.e., the bladder voided with less effort but a prompted emission rate, implying that the bladder displays an enhanced voiding efficacy after TVM. Further analyses showed that TVM postoperatively shortened the voiding time without markedly affecting the voided volume. Because the flow rate is defined by dividing the voided volume by the voiding time, an unmodified voided volume with a shortened voiding time suggests that TVM postoperatively enhanced voiding efficacy.

On the other hand, although the detailed mechanism and meaning are unclear, in POP patients whose preoperative PVA demonstrated residual volume after voiding, TVM consistently and significantly reduced the post-voided residual volume after surgery. Moreover, in the patients who were preoperatively able to empty their bladder completely, TVM did not bring about postoperative residual volume. In addition to demonstrating that TVM benefits POP patients because lasting voiding insufficiency could be detrimental as the subsequent urine retention could result in frequent and/or recurrent urinary tract infections [4], these several lines of evidence collectively suggest that TVM-diminished voiding resistance, on the one hand, lessened voiding work as it reduced the pressure gradient requisite for driving urine flow. On the other hand, it prompted urine emission and thereby increased voiding efficacy.

### 4.3. If TVM Displays Compartment or Stage Specificity

Though studies have suggested that the symptoms of POP are commonly related to the anatomical compartment involved; namely, patients suffering from anterior vaginal wall prolapse might be associated with lower urinary tract symptoms, while difficulty in defecation might be related to patients displaying severe posterior vaginal prolapse [28]; however, whether POP-associated voiding dysfunctions are compartment-specific is still controversial because studies have concluded that anatomical deficits are not always consistent with the voiding difficulties accompanying prolapse [29].

On the other hand, whether the severity of voiding dysfunctions correlates to the grade of POP is also under debate. Although the causal relationship has yet to be established, studies observed that stress incontinence and overactive bladder syndrome seem more closely associated with stage I and II POP [30,31], and POP with stage III or more might cause voiding dysfunction [32]. Nevertheless, some investigations have revealed that the POP-Q grade of POP fails to correlate with the severity of voiding dysfunction in patients [31,33].

Instead of investigating whether TVM improves the prolapse stage or modifies the compartment involved, the current study was designed to explore whether TVM displays an advantage to the voiding functions, particularly the outlet resistance and the workload of voiding, in POP patients. Our results revealed that TVM effectively diminished the voiding resistance and thereby lessened the thermodynamic workload of bladder voiding. Nevertheless, whether the TVM-decreased outlet resistance and/or voiding workload specifically correlates with the improvement of prolapse grade or occurs selectively in patients displaying specific compartment(s), deficits await further study.

### 4.4. Possible Limitations in This Study

Because the design of the current study was to retrospectively analyze existing data, the findings presented in this study have inherent limitations in both internal and external validity. In addition, considering that the data in the current study were collected from a relatively small number of patients at a single medical center, potential bias in the effects of measurement of the results cannot be excluded. Moreover, in this study, we investigated the impact of TVM on the voiding function. Nevertheless, as a well-organized bladder function comprises efficient voiding and adequate storage, further studies exploring the potential effects of TVM on urine storage, such as compliance, would offer more information about bladder function other than voiding and thereby benefit clinicians in making therapeutic decisions.

## 5. Conclusions

In conclusion, the results in the current study reveal that TVM postoperatively lessened the thermodynamic work of bladder emptying by diminishing the voiding resistance, which reduced the pressure gradient required for driving urine flow. Moreover, the TVM-reduced resistance prompted urine emission to increase voiding efficacy. Together, these findings demonstrate that TVM benefits POP patients with voiding difficulties as it objectively improves voiding function.

## Figures and Tables

**Figure 1 bioengineering-10-00853-f001:**
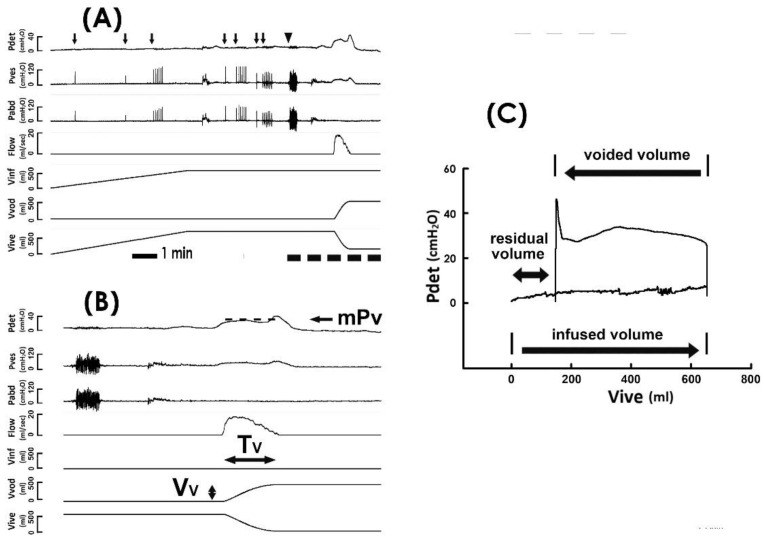
Pressure–flow and pressure–volume analyses. (**A**) Representative cystometry tracings showing the detrusor pressure (Pdet), vesical pressure (Pves), abdominal pressure (Pabd), urethral flow (Flow), infused volume (Vinf), voided volume (Vvod), and intravesical volume (Vive). Although coughs (arrow) and the Valsalva maneuver (triangle) induce marked fluctuations in Pves and Pabd, they exhibit a trivial effect on Pdet. Tracings marked by the dashed line at the bottom are displayed using a faster time base below. (**B**) Derived parameters include the mean voiding pressure (mPv), voiding time (Tv), and voided volume (Vv). Dash line indicates the level of mPv. (**C**) Pressure–volume analysis established by plotting Pdet against Vive. The trajectory of pressure–volume data shapes an enclosed loop representing a voiding cycle. The volume difference between the infused volume (the lower arrow) and voided volume (the upper arrow) denotes the residual volume (the double arrow).

**Figure 2 bioengineering-10-00853-f002:**
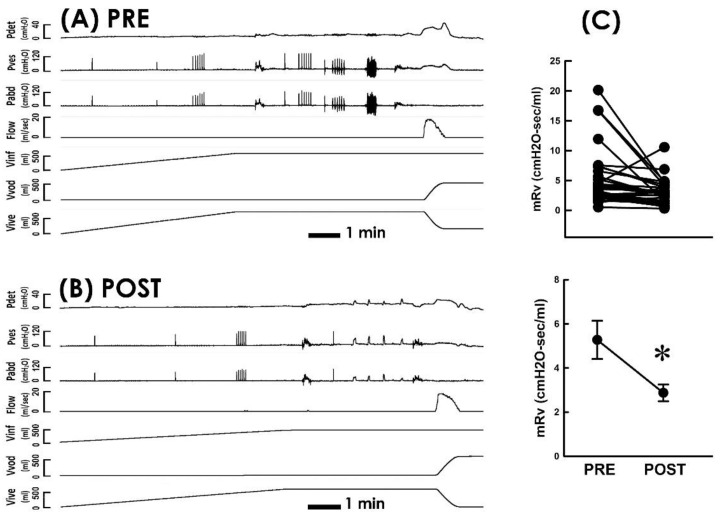
Pressure–flow studies before and after TVM (**A**,**B**). Representative cystometry of a patient with POP measured pre and postoperatively (PRE and POST, respectively). Pdet, detrusor pressure; Pves, vesical pressure; Pabd, abdominal pressure; Flow, urethral flow; Vinf, infused volume; Vvod, voided volume; Vive, intravesical volume. (**C**) Individual (upper) and summarized (lower) data of mean voiding resistance (mRv) in response to the TVM (* *p* < 0.05 vs. PRE; N = 31).

**Figure 3 bioengineering-10-00853-f003:**
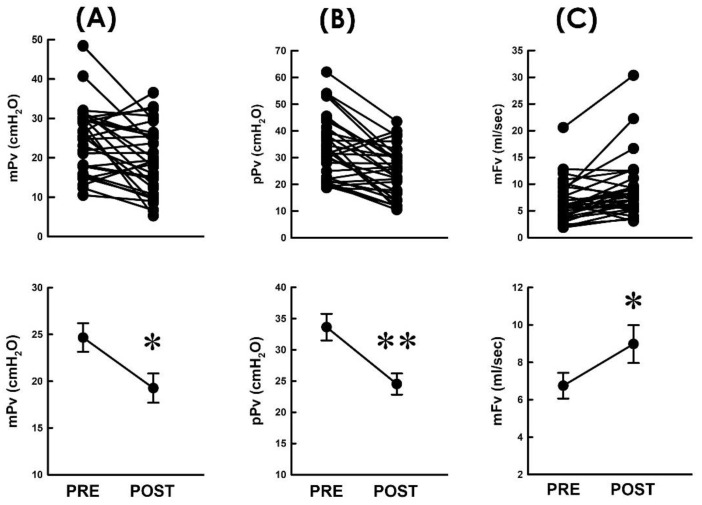
Resistance-associated parameters before and after TVM. (**A**–**C**) Individual (upper) and summarized (lower) data of the mean voiding pressure ((**A**) mPv), peak voiding pressure ((**B**) pPv), and mean voiding flow ((**C**) mFv) measured before and after TVM (PRE and POST, respectively; * *p* < 0.05, ** *p* < 0.01, vs. PRE; all N = 31).

**Figure 4 bioengineering-10-00853-f004:**
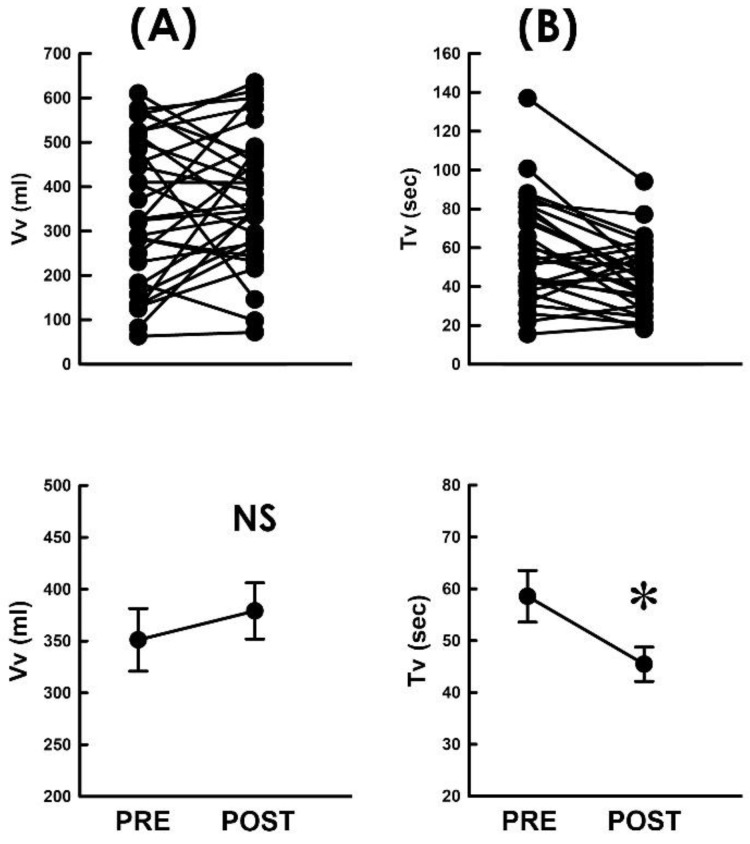
Flow-associated parameters before and after TVM (**A**,**B**). Individual (upper) and summarized (lower) data of the voided volume ((**A**) Vv) and voiding time ((**B**) Tv) measured before and after TVM (PRE and POST, respectively; NS *p* > 0.05, * *p* < 0.05 vs. PRE; both N = 31).

**Figure 5 bioengineering-10-00853-f005:**
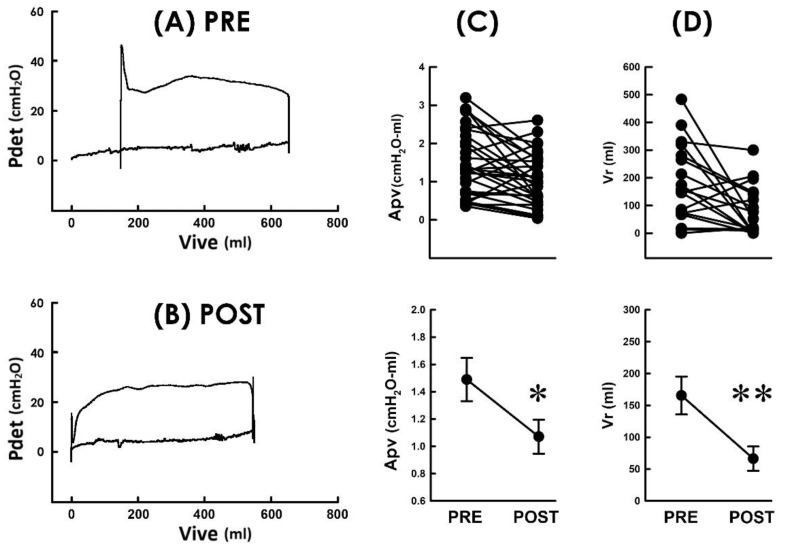
Pressure–volume loops of voiding before and after TVM (**A**,**B**). Pressure–volume analyses of a POP patient measured pre and postoperatively (PRE and POST, respectively). The TVM markedly depressed the upper boundary and shifted the loop to the left. Pdet: detrusor pressure; Vive: intravesical volume. (**C**) Individual (upper) and summarized (lower) data of the loop-enclosed area (Apv) measured before and after TVM (* *p* < 0.05 vs. PRE; N = 31). (**D**) Individual (upper) and summarized (lower) residual volume (Vr) of 21 POP patients who displayed incomplete urine emptying before surgery (** *p* < 0.01 vs. PRE; N = 21).

## Data Availability

The data that support the findings of this study are available from the corresponding author upon reasonable request.

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
