# Peer review of "Pressure-Volume Loop Analysis of Voiding Workload: An Application in Trans-Vaginal Mesh-Repaired Pelvic Organ Prolapse Patients"

_bioengineering, 2023, doi:10.3390/bioengineering10070853_

Round 1

Reviewer 1 Report

The authors have attempted to evaluated therapeutic benefits of trans-vaginal mesh (TVM) to the voiding difficulties in pelvic organ prolapse (POP) patients.

The study is methodologically well performed and is well written, however, the study population is relatively small.

 What are the limitations of this study?

Author Response

Comments of Reviewer 1:

  1. The authors have attempted to evaluated therapeutic benefits of trans-vaginal mesh (TVM) to the voiding difficulties in pelvic organ prolapse (POP) patients.

 Reply 1. We are profoundly grateful in that the reviewer has kindly spent his precious time to carefully review our manuscript and given us a lot of instructive comments/advises. All these comments have been replied point-to-point as the following. We hope our reply could comply with the reviewer’s standard.

  1. The study is methodologically well performed and is well written, however, the study population is relatively small.

Reply 2. We deep appreciate the kind and positive comment of the reviewer, and we admit that as indicated by the reviewer, the number of patient recruit in this study is relatively small; that is obviously a limitation in this study. We sincerely apologize and regret that for the patient recruitment of our project in the hospital has completed, we are not able to recruit more patients for analysis. Nevertheless, following the instructive comment of the reviewer, we have discussed this point in the revised manuscript as “considering the current study collected data of a relatively small numbers of patients from a single medical center, the potential bias in the effects of results measurement cannot be excluded.” (line 439 page 10) to faithfully demonstrate this point raised by the reviewer. We hope this modification could fairly discuss the weakness of this study and comply with the standard of the reviewer.

  1. What are the limitations of this study?

Reply 3. We completely agree and profoundly appreciate the instructive comment of the reviewer that to explain the potential limitation of this study will provide clear information about this study; and following the kind comment of the reviewer, we have added a paragraph discuss the limitations of this study as the following:

4.4 Possible limitations in this study

Because the design of the current study was to retrospectively analyze existing data, findings presented in this study have inherent limitations in both internal and external validity. In addition, considering the current study collected data of a relatively small numbers of patients from a single medical center, the potential bias in the effects of results measurement cannot be excluded. Moreover, this study investigated the impact of TVM on the voiding function. Nevertheless, as a well-organized bladder function is comprised of efficient voiding and adequate storage, further studies exploring the potential effects of TVM on urine storage, such as the compliance, would offer more information about bladder function other than voiding; and thereby benefit clinicians in making a therapeutic decision. (line 436 page 10)

We hope this paragraph could provide information about the limitation of this study and comply with the standard of the reviewer.

We profoundly appreciate all the instructive and kind comments made by the reviewer. These precious comments benefit both the revision of this manuscript and our design for future studies. 

Reviewer 2 Report

The manuscript titled: "Pressure-volume loop analysis of voiding workload in pelvic..." by Hui-Hsuan Lau and others is a pretty good piece of paper. Other than minor tweaks, I don't see any major issues with it.

Below are my comments.

1. Please correct the title - it sounds a bit strange.

  2. I would advise you to rewrite the first 2-3 sentences from the abstract, they are somewhat understandable. Similarly, the sentences from "Our results suggest..." should also be rewritten. Please underline more strongly what is the main achievement of this publication.

3. Does "workload;" this is needed as a keyword?

4. Why a period before the Introduction?

5. The introduction is at least 2 times too short, with all due respect to the authors on the Internet there is a lot to choose from when it comes to this topic. This must necessarily be supplemented.

6. Please rewrite the last paragraph so that it clearly emphasizes the purpose of the work.

Please also complete the first part of the introduction.

7. "Though USA Food and Drug Administration issued a notification regarding the long-term safety of the TVM repair in 2016..." may be some examples.

8. The quality of drawings should definitely be improved. They're a bit unfinished. Please do something with this.

9. "On the other hand, in the POP patients whose pre-operative PVA demonstrated an amount of residual volume after voiding,..." - is this the only reason.

10. "Though studies suggest stress incontinence and overactive bladder syndrome seem to be more closely associated with stage І and II of POP" - is this the only cause?

11. The summary section is completely rewritable for me, please write a decent summary.

12. Please complete citations.

To sum up, the manuscript is interesting but a bit carelessly written. I recommend Major Revision and work with fixes I would like to see.

If it is duly corrected, I will support its publication, otherwise, I will not.

Author Response

Comments of Reviewer 2:

The manuscript titled: "Pressure-volume loop analysis of voiding workload in pelvic..." by Hui-Hsuan Lau and others is a pretty good piece of paper. Other than minor tweaks, I don't see any major issues with it.

 Reply. We are deeply grateful in that the reviewer has kindly spent his precious time to carefully review our manuscript and given us a lot of instructive comments/advises. All these comments have been replied point-to-point as the following. We hope our reply could comply with the reviewer’s standard.

Comment 1. Please correct the title - it sounds a bit strange.

Reply 1. We sincerely apologize and deep appreciate that the reviewer has kindly commented the title of the previous version seems inappropriate; and following the comment, we have revised the title of the manuscript as “Pressure-volume loop analysis of voiding workload-An application in trans-vaginal mesh repaired pelvic organ prolapse patients” (line 1 page 1). We hope the revised tile could be informative and comply with the standard of the reviewer’s.

  1. I would advise you to rewrite the first 2-3 sentences from the abstract, they are somewhat understandable. Similarly, the sentences from "Our results suggest..." should also be rewritten. Please underline more strongly what is the main achievement of this publication.

Reply 2. We really appreciate that the reviewer has kindly commented that the first and the final parts of the Abstract need to be reworded.

Following the instructive comment of the reviewer we have revised the opening of the Abstract as “Though trans-vaginal mesh (TVM) offers a successful anatomical reconstruction and subjectively relieves symptoms/signs in pelvic organ prolapse (POP) patients, its objective benefits to the voiding function of the bladder have not been well established. This study investigated the therapeutic advantage of TVM on the bladder function by focusing on the thermodynamic workload of voiding.” (line 46 page 2) to explain the background of this study; and revised the last part as “Collectively, our results reveal TVM post-operatively lessened workload of bladder voiding by diminishing voiding resistance that reduced the pressure gradient requisite for driving urine flow. ” (line 63 page 2) to explain the main achievement of this study.

We hope the revised Abstract could be informative and comply with the standard of the reviewer.

  1. Does "workload;" this is needed as a keyword?

Reply 4. We completely agree that the reviewer has indicated the keyword “workload” is not appropriate. Following this comment, we have deleted the inappropriate keyword and revised the keywords as “pressure-volume analysis, thermodynamics, urethral resistance, pelvic floor reconstruction, trans-vaginal mesh repair” (line 74 page 2) in the revised manuscript. We hope these keywords could be more informative about our study.

  1. Why a period before the Introduction?

Reply 4. We sincerely apologize, as the reviewer has indicated, there is a period before the Introduction of the PDF version. We hope the reviewer could kindly forgive our fault and we have correct it in the revised version. (line 91 page 3)

  1. The introduction is at least 2 times too short, with all due respect to the authors on the Internet there is a lot to choose from when it comes to this topic. This must necessarily be supplemented.

Reply 5. We are grateful that the reviewer has kindly indicated that the Introduction of the previous version is too short and needs more literature supplement. This is also commended by the editor. Following the kind comment of the reviewer and the editor, we have extensively revised the Introduction by the following:

  1. Among therapeutic options for symptomatic POP, including consulting, observation, physical therapy, vaginal pessary, and surgical intervention (line 103 page 3).
  2. Although TVM could cause bladder damage, lower urinary tract symptoms, pelvic pain, dyspareunia [8,9], and mesh erosion [10,11] as side effects; and USA Food and Drug Administration issued a notification regarding the long-term safety of the TVM repair in 2016 (line 108 page 3)
  3. and a very recent study has demonstrated an enhance outlet resistance is associated with increased thermodynamic workload of bladder empty [21], the current study aims to explore if TVM lessens workload of bladder voiding via trimming down the voiding resistance. (line 122 page 3)

We hope this modification could improve the quality of this study and comply with the reviewer, the editor, and the journal.

  1. Please rewrite the last paragraph so that it clearly emphasizes the purpose of the work. Please also complete the first part of the introduction.

Reply 6. We deeply appreciate that the reviewer has kindly commented that it is necessary to clearly emphasize the purpose of this study in the Introduction; and the first part of the Introduction needs to be completed.

Following the comment of the reviewer we have revised the last part of the Introduction as “For this purpose, the impact of TVM on the thermodynamic work and the urethral resistance of voiding in POP patients, were analyzed by retrospectively comparing the pre- and post-operative urodynamic investigations.”(line 126 page 3).

In addition, we have revised the first part of the Introduction as “Thereby, therapies for POP have developed not only for treating bowel or sexual dysfunctions but also for remedying voiding deficits. [5].” (line 101 page 3) to complete this paragraph. We hope these modifications could improve the readability of this manuscript and be informative about the aim/purpose of this study.

  1. "Though USA Food and Drug Administration issued a notification regarding the long-term safety of the TVM repair in 2016..." may be some examples.

Reply 7. We are profoundly grateful that the reviewer has kindly encourage us to give examples for the side effects of TVM that prompt the FDA to issue a notification; and following the instructive comment of the reviewer, we have revised this paragraph as “Although TVM could cause bladder damage, lower urinary tract symptoms, pelvic pain, dyspareunia [8,9], and mesh erosion [10,11] as side effects; and USA Food and Drug Administration issued a notification regarding the long-term safety of the TVM repair in 2016 [12]” (line 108 page 3). We hope this modification could provide more information about this notification and improve the quality of this manuscript.

  1. The quality of drawings should definitely be improved. They're a bit unfinished. Please do something with this.

Reply 8. We sincerely apologize and hope the reviewer could kindly forgive the quality of drawings in the PDF file is too low that definitely needs to be improved. We have also found this problem which could be a reduced resolution in figure during converting a DOC file into PDD file.

In accompanied with the revised manuscript, we have upload figure files of high resolution, we hope this problem can be overcome, and once again apologize for this problem.

  1. "On the other hand, in the POP patients whose pre-operative PVA demonstrated an amount of residual volume after voiding,..." - is this the only reason.

Reply 9. We admit that as indicated by the reviewer, though we proposed the volume difference between the infused and the voided volumes was the post-voided residual volume; and it was post-operatively decreased by TVM, the precise meaning of this volume and its response to TVM is unclear so far. Therefore, following the comment of the reviewer, we have revised the indicated paragraph as ” On the other hand, though the detail mechanism and meaning is so far unclear, in the POP patients whose pre-operative PVA demonstrated an amount of residual volume after voiding, TVM consistently and significantly reduced the post-voided residual volume after surgery.” (line 392 page 9), and we hope the revised version could more faithfully explain our results in this study and comply with the standard of the reviewer.

  1. "Though studies suggest stress incontinence and overactive bladder syndrome seem to be more closely associated with stage І and II of POP" - is this the only cause?

Reply 10. We completely agree the comment of the reviewer that though the cited publication has suggested stress incontinence and overactive bladder syndrome seem to be more closely associated with stage І and II of POP, the detail causal relationship between POP and stress incontinence/overactive bladder has not been well established. Following the instructive comment of the reviewer, we have revised this paragraph as “Though the causal relationship is so far yet been established, studies observed that stress incontinence and overactive bladder syndrome seem to more closely associated with stage І and II of POP [30,31],” (line 417 page 9), and we hope the revised version could more faithfully discuss this reference and provide clear information about our study.

  1. The summary section is completely rewritable for me, please write a decent summary.

Reply 11. We apologize sincerely and hope the reviewer could kindly forgive that the quality of Conclusion in the previous version fails to comply with the standard of the reviewer. Following the comment of the reviewer, we have extensively revised the Conclusion of our manuscript as the following:

In conclusion, results in the current study reveal that TVM post-operatively lessened the thermodynamic work of bladder emptying via diminishing the voiding resistance that reduced the pressure gradient requisite for driving urine flow. Moreover, the TVM-reduced resistance prompted urine emission to increase voiding efficacy. Together, these findings demonstrated that TVM benefits POP patients with voiding difficulties as it objectively improves the voiding function. (line 451 page 10)

We hope the quality of the Conclusion of the revised manuscript could be improved

  1. Please complete citations.

Reply 12. We are really grateful in the kind advise of the reviewer; and following the advice, we have cited the added literatures in the Reference of the revised version as reference 5, 8-11, and 21. We deeply appreciate the kind help of the reviewer.

To sum up, the manuscript is interesting but a bit carelessly written. I recommend Major Revision and work with fixes I would like to see. If it is duly corrected, I will support its publication, otherwise, I will not.

We deep appreciate in the well-intended and instructive comments made by the reviewer. These precious comments benefit both the revision of this manuscript and our design for future studies. We hope all the modifications we made could comply with the standard of the reviewer and the journal.

Round 2

Reviewer 2 Report

The authors very good improved this manuscript. I think that this manuscript is suitable for publication.